# Antibiotic Use for Sepsis in Hospitalized Neonates in Botswana: Factors Associated with Guideline-Divergent Prescribing

**DOI:** 10.3390/microorganisms11112641

**Published:** 2023-10-27

**Authors:** Jameson Dowling, Tonya Arscott-Mills, One Bayani, Mickael Boustany, Banno Moorad, Melissa Richard-Greenblatt, Nametso Tlhako, Morgan Zalot, Andrew P. Steenhoff, Alemayehu M. Gezmu, Britt Nakstad, Jonathan Strysko, Susan E. Coffin, Carolyn McGann

**Affiliations:** 1College of Public Health, Temple University, Philadelphia, PA 19122, USA; dowlingj@chop.edu; 2Division of Infectious Diseases, Children’s Hospital of Philadelphia, Philadelphia, PA 19104, USAcoffin@chop.edu (S.E.C.); 3Department of Pediatrics, Wake Forest School of Medicine, Winston Salem, NC 27101, USA; 4Faculties of Medicine & Health Sciences, Department of Paediatric & Adolescent Health, University of Botswana, Gaborone P.O. Box 00701, Botswana; 5Botswana-UPenn Partnership, University of Pennsylvania & University of Botswana, Gaborone P.O. Box 45498, Botswana; 6Department of Laboratory Medicine and Pathobiology, University of Toronto, Toronto, ON M5S 1A1, Canada; 7Public Health Ontario, Toronto, ON M5G 1M1, Canada; 8Perelman School of Medicine, University of Pennsylvania, Philadelphia, PA 19104, USA; 9Department of Pediatrics, Children’s Hospital of Philadelphia, Philadelphia, PA 19104, USA; 10Division of Neonatology, Children’s Hospital of Philadelphia, Philadelphia, PA 19104, USA

**Keywords:** neonatal sepsis, antibiotic utilization, antibiotic guidelines, antimicrobial resistance, middle-income country, global health, neonatology, infectious diseases

## Abstract

In low- and middle-income countries, where antimicrobial access may be erratic and neonatal sepsis pathogens are frequently multidrug-resistant, empiric antibiotic prescribing practices may diverge from the World Health Organization (WHO) guidelines. This study examined antibiotic prescribing for neonatal sepsis at a tertiary referral hospital neonatal unit in Gaborone, Botswana, using data from a prospective cohort of 467 neonates. We reviewed antibiotic prescriptions for the first episode of suspected sepsis, categorized as early-onset (EOS, days 0–3) or late-onset (LOS, >3 days). The WHO prescribing guidelines were used to determine whether antibiotics were “guideline-synchronous” or “guideline-divergent”. Logistic regression models examined independent associations between the time of neonatal sepsis onset and estimated gestational age (EGA) with guideline-divergent antibiotic use. The majority (325/470, 69%) were prescribed one or more antibiotics, and 31 (10%) received guideline-divergent antibiotics. Risk factors for guideline-divergent prescribing included neonates with LOS, compared to EOS (aOR [95% CI]: 4.89 (1.81, 12.57)). Prematurity was a risk factor for guideline-divergent prescribing. Every 1-week decrease in EGA resulted in 11% increased odds of guideline-divergent antibiotics (OR [95% CI]: 0.89 (0.81, 0.97)). Premature infants with LOS had higher odds of guideline-divergent prescribing. Studies are needed to define the causes of this differential rate of guideline-divergent prescribing to guide future interventions.

## 1. Introduction

Neonatal sepsis is a major cause of neonatal mortality and morbidity globally, accounting for 500,000 to 900,000 neonatal deaths annually [1,2]. These deaths are disproportionately concentrated in low- and middle-income countries (LMIC) [2,3,4].

Antimicrobial-resistant (AMR) pathogens are becoming an increasingly common cause of fatal neonatal sepsis; an estimated 214,000 neonatal deaths per year are attributable to infections caused by AMR pathogens [5]. The highest burden of deaths attributable to AMR organisms is observed in sub-Saharan Africa [6].

Neonatal units have high rates of antibiotic usage [7,8,9], much of which is considered inappropriate [10,11]. Inappropriate empirical antibiotic use has been associated with increased rates of neonatal mortality and adverse long-term outcomes [7,12]. The relationship between inappropriate antibiotic use and AMR [10,11,13,14] is complex. A rising rate of AMR likely drives increased prescribing of broad empiric antibiotics, a practice that can further AMR but may be necessary when treating patients in a setting with highly resistant organisms [15,16].

Empiric antibiotic treatment guidelines are a key tool designed to improve the quality of antibiotic prescribing by preventing “bug-drug mismatches” and reducing the unnecessary prescribing of antibiotics [17]. However, many guidelines, including those developed by the World Health Organization (WHO), were informed primarily from data in high-income settings, where the leading cause of neonatal sepsis is infection with Group B *Streptococcus* [18,19]. In LMIC, Gram-negative pathogens are responsible for most early-and late-onset sepsis, and these organisms harbor the highest rates of AMR [2,13,20,21]. Thus, the WHO antibiotic guidelines may be less useful in resource-limited settings [14]. Synchrony with treatment guidelines may be affected by access to appropriate regimens [22], limited laboratory capacity to conduct antimicrobial sensitivity testing [23], as well as other system factors [22]. Prior work has highlighted that the time of sepsis onset [24,25,26] and gestational age [1,22,27,28,29] influence antibiotic use for neonatal sepsis; however, previous research has focused mainly on early-onset sepsis.

This study examined antibiotic use in a neonatal cohort from a tertiary referral hospital neonatal unit in Gaborone, Botswana, to identify patient-level factors associated with guideline-divergent prescribing compared to the existing WHO guidelines [30]. Our work extends that of Mudzikati and Kitt, who found that 10% of patients in this Botswanan NICU develop laboratory-confirmed sepsis [20] and that this was associated with 23% mortality [3]. Our findings identify specific scenarios with an increased risk of guideline-divergent care and suggest that additional work is needed to better define contemporary empiric treatment regimens and enhance guideline-congruent prescribing for neonatal sepsis.

## 2. Materials and Methods

### 2.1. Setting and Population

The study setting was a 530-bed tertiary public referral hospital in Gaborone, Botswana, where over 8000 deliveries occur annually. The 36-bed neonatal unit is divided into five rooms that provide varying levels of acuity and support. Neonatal care in this unit includes supplemental oxygen and mechanical ventilatory support, cardio–respiratory monitoring, enteral and parenteral nutrition, thermoregulation, phototherapy, and fluid and electrolyte supplementation.

### 2.2. Study Cohort

Data from a prospective cohort of 467 neonates admitted to the referral hospital’s neonatal unit from November 2020 to December 2021 were utilized for this study. Approximately 75% of admitted neonates were born at the study hospital. Neonates aged < 96 h at admission were eligible for the cohort; neonates whose mothers had a maternal age of <18 years or a maternal diagnosis of severe acute respiratory syndrome coronavirus 2 (SARS-CoV-2) within the past 10 days were excluded. This study used a sub-cohort of 325 neonates prescribed antibacterial antimicrobials. The full cohort was approved by the institutional review boards of the institution, Botswana Ministry of Health Research and Development Committee, University of Botswana, the University of Pennsylvania, the hospital IRB and Children’s Hospital of Philadelphia.

### 2.3. Study Data, Outcome, and Covariates

Maternal epidemiologic data was assessed by chart review whereas the mother was interviewed to determine elements of her past medical history and past medication use. Neonates were followed until neonatal unit discharge or death. Study personnel reviewed the neonates’ charts to obtain information on birth history and parameters and daily exposures including antibiotic administration. Antibiotic prescriptions for the first episode of suspected sepsis were reviewed. The primary outcome was defined as “guideline-divergent” or “guideline-synchronous” based on the 2013 WHO neonatal sepsis antibiotic guidelines. First-line antibiotics, per the WHO guidelines, are gentamicin with ampicillin or benzylpenicillin [30]. Two main exposure variables were investigated: (1) time of neonatal sepsis, classified as either early-onset sepsis (EOS) if occurring in the first 3 days of life or late-onset sepsis (LOS) neonatal sepsis if occurring after the first 3 days of life at the time that antibiotics were first prescribed [31,32]; and (2) estimated gestational age (EGA) in weeks. Although early-onset sepsis may be defined as up to 7 days, for continuously hospitalized patients, day of life 0–3 is often used to assess differences related to hospital-based exposures [20,31,32,33,34,35,36,37,38]. To address the differences in sepsis definitions, we evaluated the day of life of sepsis onset as a continuous variable for additional insight. EGA in weeks was a continuous variable measured via the date of the last menstrual period or Ballard scoring, a common clinical method of determining gestational age [39] when the date of the last menstrual period was unknown. For additional insight, EGA was defined categorically as <28 weeks, 28–31 weeks, 32–36 weeks, or ≥37 weeks. Other clinically important variables were assessed for confounding. Blood cultures were collected at the discretion of the clinical team based on common clinical criteria [40,41]. Blood culture collection was recorded in a laboratory log. Blood cultures underwent incubation using an automated system (BACT/ALERT®, BioMérieux, Marcy-l’Etoile, France), and bacterial identification was done manually using biochemical tests and antimicrobial sensitivity testing using phenotypic (disc diffusion) methods. Discharge status was defined based on a review of the unit admissions and discharge book.

### 2.4. Analysis

Statistical analysis was conducted using R (V 4.2.3, R Core Team,, Vienna, Austria). Descriptive analyses were conducted, with categorical variables described using total counts and frequencies and continuous variables described using medians and interquartile ranges (IQR). The normality of each continuous variable was assessed. The amount of data missing for each variable under study was evaluated, with a cutoff of 10% missingness. Bivariate analysis was conducted to detect significant differences at the 0.05 significance level via Fisher’s exact tests for categorical variables and the Wilcoxon ranked sum test for continuous variables since all were abnormal. The other bivariate analysis results were used to assess for potential confounding.

Multivariate analysis was then conducted with logistic regression models to detect any significant differences at the 0.05 significance level. Based on bivariate analyses and conceptual framework, EGA was controlled for in the model of the time of neonatal sepsis onset and the guideline synchrony of the antibiotic. For the model of EGA and guideline-synchrony of antibiotic use, it was determined that the variables under consideration were mediators, not potential confounders, and thus, no variables were controlled for in this model.

## 3. Results

### 3.1. Characteristics of Cohort

In this prospective cohort of 467 neonates admitted to the neonatal unit between November 2020 and December 2021, we examined the use of antibacterial and antimicrobial agents prescribed for 325 neonates during their first episode of suspected sepsis. In our cohort, the overall median EGA was 34 weeks (IQR 30–38), and 81 (25%) neonates had perinatal exposure to HIV. The in-hospital mortality rate was 11% (35/325) (Table 1). Five (14%) of the thirty-five neonates were prescribed guideline-divergent antibiotics (d The median age at the time of neonatal sepsis onset was 1 day (IQR 1–3); 293 (90%) neonates had EOS. In our cohort, there were 25 episodes of laboratory-confirmed sepsis, of which 9 (36%) were resistant to the WHO guideline first-line antibiotics (Table A1, Appendix A). Of the 12 pathogens isolated from patients with EOS, 3 (25) were resistant to the WHO first-line antibiotics, compared to 6 (46%) of the 13 pathogens isolated from patients with LOS (Table A1, Appendix A).

### 3.2. Guideline-Divergent Antibiotic Use

Guideline-divergent antibiotics were prescribed for 10% (31/325) of neonates treated empirically for sepsis. The use of guideline-divergent antibiotics was more common in neonates treated for LOS than in those with EOS (25% vs. 8%, *p* = 0.0057). The median day of life at the time of sepsis onset was similar for neonates with EOS who were prescribed guideline-divergent antibiotics compared to guideline-synchronous antibiotics. However, among neonates with LOS, the median day of life at the time of neonatal sepsis onset of neonates prescribed guideline-divergent antibiotics was nine (IQR 5–17.75) compared to four (IQR 4–4) in those prescribed guideline-synchronous antibiotics (Table 2).

The guideline-divergent antibiotic regimens prescribed for the treatment of LOS included amikacin, piperacillin–tazobactam, vancomycin, cefotaxime, or meropenem, while regimens used for EOS were primarily ampicillin only (Table A2, Appendix A).

Neonates prescribed guideline-divergent antibiotics had a lower median EGA (31 weeks) as compared to neonates prescribed guideline-synchronous antibiotics (34 weeks; *p* < 0.001). Neonates with an EGA < 28 weeks had the highest proportion of guideline-divergent antibiotic use. Most neonates treated for sepsis had a blood culture collected (216/325; 66%). There was no difference in the rate of blood culture collection between those prescribed guideline-synchronous versus guideline-divergent antibiotics (66% vs. 71%; *p* = 0.6908) (Table 2).

In bivariate analysis, the likelihood of guideline-divergent prescribing was associated with the duration of neonatal unit stay, maternal age, and birth weight. There was no difference in the likelihood of guideline-divergent prescribing based on infant sex, HIV exposure status, delivery mode, or discharge status.

The use of guideline-divergent antibiotics was not associated with in-hospital death (*p* = 0.3551); the mortality rate was 16% in neonates prescribed guideline-divergent antibiotics compared to 10% in neonates prescribed guideline-synchronous antibiotics.

### 3.3. Independent Factors Associated with Guideline-Divergent Use of Antibiotics

After controlling for EGA, neonates with LOS had 4.89 times the odds of guideline-divergent antibiotic use compared with neonates with EOS (95% CI 1.81–12.57) (Table 3). When we examined the time of sepsis onset as a continuous variable, we found that every 1-day increase in the day of life at the time of neonatal sepsis onset resulted in 37% increased odds of guideline-divergent antibiotic use after controlling for EGA (aOR 1.37 [95% CI, 1.17–1.69]). Every 1-week decrease in EGA resulted in 11% increased odds of guideline-divergent antibiotic use (OR 0.89 [95% CI 0.81–0.97]) (Table 3).

## 4. Discussion

In this cohort of hospitalized neonates in a tertiary hospital in Botswana, a middle-income country, we found a relatively low rate (10%) of guideline-divergent antibiotic use based on first-line prescribed antibiotics. However, specific groups of neonates, those with low EGA and LOS, had higher odds of treatment with guideline-divergent antibiotics, which was aligned with our observation that nearly 40% of organisms isolated from blood cultures were resistant to guideline antibiotics.

Previous studies evaluating guideline-divergent antibiotic use for neonatal sepsis, were limited to EOS [24,25,26]. The present study evaluated both EOS and LOS, as well as the impact of day of life as a continuous variable. Older neonates and those with LOS had higher odds of guideline-divergent antibiotic use. This may be explained by treatment decisions for LOS more often being based on clinical findings [41] as opposed to EOS, which is often empiric treatment based on risk factors [8,42]. Additionally, LOS is more likely to be caused by hospital-acquired pathogens that have high rates of AMR [14,43] and provider knowledge of this and of local patterns of resistance may also influence choice of antibiotic. Changing epidemiology and continual increases of AMR rates prompt an ongoing need for research.

Prematurity is a major risk factor for sepsis [1,27,28,41], in part due to increased length of stay, increased invasive device use, and immature immunity; however, limited data exists evaluating the association of gestational age with guideline-divergent antibiotic use for neonatal sepsis [22]. Prior work highlights an association between low EGA and neonatal sepsis due to AMR pathogens [44], which may influence guideline-divergent antibiotic use for neonatal sepsis in this high-risk group. In this study, neonates with lower EGA had higher odds of guideline-divergent antibiotic use. Knowledge of prematurity as a risk factor and the association with AMR could affect clinicians’ choice of antibiotics.

Guideline-divergent prescribing has been categorized as “inappropriate” in some settings, and certainly, inappropriate prescribing is occurring in LMICs, where antimicrobial stewardship oversight is often inadequate. However, the associations we identified of time of neonatal sepsis onset and EGA with guideline divergence likely reflect a thoughtful risk calculation on the part of the prescribing clinician, one that incorporates available clinical and laboratory data and a rising concern for nosocomial sepsis due to multidrug-resistant pathogens with each hospital day. Given the existing critiques of the WHO guidelines as being under-representative of LMIC settings and growing AMR, these identified associations may be incorporated into future antibiotic-prescribing algorithms tailored to LMIC settings, which consider both local epidemiology and clinical risk factors.

The present study had limitations. The small sample size may have limited the detection of significant effects of certain variables, resulting in the exclusion of these variables as controlling factors in regression analysis. A lack of data prevented us from examining the frequency of guideline-divergent dosing or duration of antibiotics. Similarly, we are unable to comment on provider rationale for guideline-divergent antibiotic choice. Local system factors, such as antibiotic shortages, concurrent outbreaks of multidrug-resistant infections, and prescriber details, may have influenced clinicians’ decisions when prescribing antibiotics for neonatal sepsis, but data on these are either not collected or were not available. Additional patient-level factors and perinatal history that could have been accounted for, including signs and symptoms at initial presentation or leading to sepsis evaluation, potentially introduced residual confounding. Likewise, maternal factors and symptoms were not available. This study was also impacted by intermittent enrollment and data collection pauses due to the COVID-19 pandemic, possibly introducing selection bias. Lastly, due to the complex drivers of neonatal mortality, we have insufficient power to examine the independent impact of guideline-divergent prescribing on mortality or other patient outcomes.

Antimicrobial resistance significantly impacts neonatal sepsis patients’ outcomes and influences, as well as results from, guideline-divergent antibiotic use. As suggested by other groups [6,11,12,13,14,20,22,28,43], our findings of guideline-divergent antibiotic use among neonates at increased risk of neonatal sepsis due antibiotic-resistant pathogens could indicate that the treatment guidelines no longer align with contemporary AMR patterns in many parts of the globe. In 2015 in South Africa, it was found that guideline antibiotics are not appropriate for 63% of causative pathogens based on antibiotic susceptibilities [10]. Future studies are needed to refine our understanding of other drivers of guideline-divergent antibiotic prescribing in the neonatal population including systems-level factors which may help guide future interventions. 

In conclusion, we observed an association between time of neonatal sepsis onset and lower EGA with increased use of guideline-divergent antibiotic regimens for neonatal sepsis. Local factors, such as antibiotic shortages and rates of multidrug resistant infections prevalent in the neonatal unit, may have influenced the decisions made by healthcare providers when prescribing antibiotics. Since the time of this study, local antibiotic guidelines have been published to allow for adapting to the hospital antibiogram [45], however recently updated WHO guidelines have not changed [18]. Our observations add to the existing literature and knowledge by providing insight on the burden of and key clinical factors associated with guideline-divergent antibiotic use for neonatal sepsis, especially in LMIC. Our findings can help inform local antibiotic practices and highlight the importance of capacity building in laboratory diagnostics and antimicrobial stewardship. Further research is needed to identify additional drivers, inform targeted interventions in similar settings, and guide more appropriate neonatal sepsis antibiotic guidelines. 

## Figures and Tables

**Table 1 microorganisms-11-02641-t001:** Characteristics of neonates treated with antibiotics.

Covariate	Study Sample (N = 325)
Median EGA ^1^, weeks (IQR)	34 (30–38)
Median Birthweight, grams (IQR)	1795 (1328–2825)
Infant Sex	
Female	167 (52%)
Male	153 (48%)
HIV Exposure	
Yes	81 (25%)
No	244 (75%)
Median Duration of Stay ^2^, days (IQR)	8 (4–16)
Median Maternal Age, years (IQR)	29 (24–35)
Delivery Mode	
Vaginal	223 (69%)
Cesarean section	101 (31%)
Discharge Status	
Alive	290 (89%)
Deceased	35 (11%)

^1^ EGA, estimated gestational age, ^2^ Duration of stay in neonatal unit.

**Table 2 microorganisms-11-02641-t002:** Patient factors associated with guideline-divergent use of antibiotics.

Exposure	Antibiotic Utilization		*p*-Value ^1^
	Guideline- Divergent Use *n* = 31 *n*	Guideline- Synchronous Use *n* = 294 *n*	
Time of Neonatal Sepsis Onset EOS ^2^ LOS ^3^ Day of Life at Sepsis Onset All episodes, median (IQR) EOS, median (IQR) LOS, median (IQR)			
	23 (8%)	270 (92%)	0.0057
	8 (25%)	24 (75%)	
	2 (1–3.5)	1 (1–2)	<0.001
	1 (1–2.5)	1 (1–2)	
	9 (5–17.75)	4 (4–4)	
EGA Median (IQR) <28 weeks 28–31 weeks 32–36 weeks ≥37 weeks			
	31 (29–34)	34 (31–38)	<0.001
	6 (23%)	20 (77%)	
	10 (13%)	69 (87%)	
	9 (8%)	105 (92%)	
	6 (6%)	96 (94%)	
Blood Culture Collected Yes No			
	22 (10%)	194 (90%)	0.6908
	9 (8%)	100 (92%)	

^1^ Fisher’s exact test was used for categorical variables since guideline-divergent use was rare. The Wilcoxon rank sum test was used for continuous variables since all variables were not normally distributed. ^2^ EOS, early-onset sepsis (day of life 0–3). ^3^ LOS, late-onset sepsis (day of life > 3).

**Table 3 microorganisms-11-02641-t003:** Association between time of neonatal sepsis onset and EGA with guideline-divergent use of antibiotics (N = 325).

Exposure	Antibiotic Utilization			
	OR (95% CI)	*p*-Value	aOR ^1^ (95% CI)	*p*-Value
Time of Neonatal Sepsis Onset EOS (*n* = 294) LOS (*n* = 31)				

	1.00	0.0032	1.00	0.0011
	3.91 (1.51, 9.94)		4.89 (1.81, 12.57)	
EGA Continuous ^2^				
	0.89 (0.81, 0.97)	0.0088	-	-

^1^ aOR = adjusted odds ratio; time of neonatal sepsis onset model was adjusted for EGA. ^2^ Odds ratio for continuous EGA model represents the odds of receiving guideline-divergent antibiotic use for every 1-week increase in day of life.

## Data Availability

The data presented in this study are available upon request from the corresponding author. The data are not publicly available due to privacy issues and rules of data sharing as part of the participant consent process.

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
