# Peer review of "Antibiotic Use for Sepsis in Hospitalized Neonates in Botswana: Factors Associated with Guideline-Divergent Prescribing"

_microorganisms, 2023, doi:10.3390/microorganisms11112641_

Round 1
Reviewer 1 Report
In this manuscript, Dowling et al., examined antibiotic prescription for neonatal sepsis in a neonatal unit in Botswana. Overall, the study is well performed and the manuscript can be accepted in its present form.
In this study authors, have determined the association between the time of neonatal sepsis onset (early or late) and lower estimated gestational age with divergent antibiotic regimens as per WHO guidelines. The study suggested that treatment guidelines does not align with contemporary antimicrobial resistance.
The sample size used for the study is low but the authors have already mentioned about this limitation in the manuscript.Tables and figures are fine as per my understanding.
Reviewer 2 Report
work that is applicable in many low- and middle-income countries, where antimicrobial access may be erratic and neonatal sepsis pathogens are frequently multidrug-resistant, empiric antibiotic prescribing practices may diverge from World Health Organization (WHO) guidelines. Unfortunately, this is an established practice in those areas and this paper gives a good account of it. It is written clearly, comprehensively and interestingly for readers.
Reviewer 3 Report
It is not clear to me how the authors explain guideline diverted antibiotic therapy in the early onset sepsis day 1-3.
what means laboratory co firmed sepsis ( cultures, biomarkers other) if only 26 episodes of sepsis were confirmed how the adequacy of treatment was confirmed.
The work is interesting since it enriches the information about infections and mortality in early onset sepsis in newborns in their first 3 days of life. It shows as well that in the case of late onset sepsis ( day 4-9) in low income country in newborns ( day 4-9) mortality is 16%. Empiric therapy based on microprophile is effective .